# Small Changes in Patient Arrival and Consultation Times Have Large Effects on Patients’ Waiting Times: Simulation Analyses for Primary Care

**DOI:** 10.3390/ijerph20031767

**Published:** 2023-01-18

**Authors:** Matthias Grot, Simon Kugai, Lukas Degen, Isabel Wiemer, Brigitte Werners, Birgitta M. Weltermann

**Affiliations:** 1Institute of Management, Operations Research, Ruhr University Bochum, Universitätsstr. 150, 44801 Bochum, Germany; 2Institute for General Practice and Family Medicine, University Hospital Bonn, Venusberg-Campus 1, 53127 Bonn, Germany

**Keywords:** general practice, appointment scheduling, consultation strategies, waiting times, stochastic discrete event simulation

## Abstract

(1) Background: Workflows are a daily challenge in general practices. The desired smooth work processes and patient flows are not easy to achieve. This study uses an operational research approach to illustrate the general effects of patient arrival and consultation times on waiting times. (2) Methods: Stochastic simulations were used to model complex daily workflows of general practice. Following classical queuing models, patient arrivals, queuing discipline, and physician consultation times are three key factors influencing work processes. (3) Results: In the first scenario, with patients arriving every 7.6 min and random consultation times, the individual patients’ maximum waiting time increased to more than 200 min. The second scenario with random patient arrivals and random consultation times increased the average waiting time by up to 30 min compared to patients arriving on schedule. A busy morning session based on the second scenario was investigated to compare two alternative intervention strategies to reduce subsequent waiting times. Both could reduce waiting times by a multiple for each minute of reduced consultation time. (4) Conclusions: Aiming to improve family physicians’ awareness of strategies for improving workflows, this simulation study illustrates the effects of strategies that address consultation times and patient arrivals.

## 1. Introduction

Workflows remain a daily challenge in general practices. Poor workflows promote direct waiting times, productivity losses, dissatisfaction, and no-show rates on behalf of patients, as well as chronic stress among practice personnel [1,2,3,4,5]. While indirect waiting times represent the willingness to wait for a prescheduled appointment [6,7], direct waiting times represent a patient’s time in the practice waiting for the consultation [7]. According to German data from 2008 to 2020, about 90% of patients experience some direct waiting time before consulting their physician. Waiting times range from no waiting time to longer than 120 min, with outliers of more than 2 h in up to 2% of patients [8]. High waiting times are associated with less patient satisfaction [4,5] and medical compliance, as well as increased no-show rates [3]. Hence, the investigation of direct waiting times is crucial for improving medical care. In recent years, simulation models and other studies from operations research have offered profound theoretical insights into how to optimize workflows with relevant practical implications [9,10,11,12].

The theoretical foundation of research on waiting times and workflows is based on the queuing theory [10,11], which refers to the mathematical study of an operating system with waiting lines. The main purpose is to gain information on queue lengths, the waiting times of clients requiring some kind of service, and the utilization of the service system. A theoretical general practice can be considered as a queuing system with three main components (see Figure 1): 1. the arrival process (frequency and types of patients arriving), 2. the service mechanism (here: physician capacities, including consultation time and idle time before serving the next patient), and 3. the queuing discipline (treatment order, e.g., with/without unscheduled emergency patients).

The mathematical results from operations research, which are well published, have provided the following fundamental principles [12,13]: A fully utilized server (doctor) will lead to infinite waiting times in the long run, even if there is only a small degree of variation in arrival or consultation time. Under an equidistant schedule (e.g., every 10 min), waiting times per patient will always increase the later a patient is scheduled (even in optimized schedules) if there is only little variation [14]. Additional sources of variation that play a key role in scheduling are no-shows and walk-ins without appointments [15]. To minimize waiting times, the idle time of the physician needs to increase [16]. The latter principle is especially important in primary care with its patient-centered care philosophy [17]; frequently, a single patient’s needs are hardly predictable when scheduling [18].

In operations research, various models were developed in order to find the appointment schedule that performed best in terms of patient waiting times, physician idle times, or other performance metrics in clinical environments [15,19]. Regarding the first aspect of a practice´s queuing system, the patient arrival process (see Figure 1), the main challenges for practices are the allocation of appointments, the distribution of patient requests to weekdays, the number of walk-ins without appointments, appropriate time slots, and suitable buffers [20,21,22,23]. The second aspect of the model, the physician capacity can be influenced by increasing the number of physicians or increasing the capacity of each physician. Theoretically, the latter can be achieved with shortened consultation times, reduced set-up times, and more equal consultation lengths using, e.g., standardized information and consultation procedures, yet patients´ complex care demands limits to the implementation of these options. Furthermore, the influence of patient scheduling and physician time on the quality of care needs to be considered. A positive example from the literature addressed systematic diabetic care scheduling: A decrease in consultation time with an increase in patients seen per day allowed the physician to see more patients before disease deterioration, which yielded better diabetes control for all patients affected by a higher glucose level [24].

This paper aims to support the interdisciplinary transfer of knowledge from the appointment scheduling and queuing literature to family medicine. Using data from the literature and appropriate assumptions, we present two scenarios to illustrate the general effects of patient arrival and consultation times on waiting times. To reflect a frequent scenario in general practices, a single busy morning is analyzed while using two additional actions that demonstrate the benefits of responding quickly to strongly increasing waiting times.

## 2. Materials and Methods

### 2.1. Simulation Model

To evaluate the impact of patients’ arrival rates and physician consultation times on waiting times, we implemented a simulation model by using the commercial software Anylogic^®^ (The AnyLogic Company, Chicago, IL, USA). If the next patient always arrives as soon as the treatment of the previous one is completed, there are no waiting times. This is completely different when patient arrivals and treatment durations fluctuate. It is well known from queuing theory that for an almost fully utilized server with varying arrivals, the queue length tends to infinity. The simulation model illustrates a simple treatment process to address the overall impact of variation in consultation time (scenario 1) and patient arrivals (scenario 2). These are realistic scenarios because care processes may vary in content and include additional workflows, e.g., blood collection, ECG, or lung function testing, which differ by patient and practice. The patient flow in a practice is modeled as a single-server queuing system to demonstrate the general effects on direct waiting times. The patient flow is as follows: Initially, the patient arrives at the practice. If the physician is idle, the patient directly enters the service. A patient who finds the physician busy joins the end of a single waiting queue. When the physician completes a service, the next patient is chosen from the waiting queue on a first-come-first-served (FCFS) basis. No priorities are considered. After consultation, the patient immediately leaves the practice. All patients generated for a particular day have to be treated on the same day. If the work capacity limit of the physician is reached, extra hours are conducted.

### 2.2. System Parameters and General Assumptions

The input parameters of our simulation model are based on the literature and the experiences of primary care researchers and general practitioners. On each day, patients are treated for eight hours, which are separated into a five-hour morning and three-hour afternoon session, interrupted by a two-hour break. When consultation times exceed the five-hour morning session, the physician sees patients during the break and works overtime. The simulation covers a period of one year consisting of 260 independently simulated working days, which do not differ in patient arrival or physician consultation patterns. Therefore, the results of the simulation are the average of 260 runs, each with about 63 patients. The physician consultation times follow the commonly used log-normal distribution [25,26] to represent typical variations in the duration of treatment.

### 2.3. Scenarios Addressing the Arrival Process (Scenarios 1 and 2)

For patient arrivals, two cases are considered. First, we analyze the effects when patient arrivals are fixed, accounting for situations when patients meet their appointments on time. Second, simulating walk-ins without appointments, patients are generated following the commonly used Poisson process with exponentially distributed inter-arrival times [27,28]. The average consultation and average patient inter-arrival times are set to 7.6 min based on the empirical service time of general practitioners in Germany, which is the shortest consultation time compared to five other European countries, with an overall mean of 10.7 min [29]. Regarding the consultation times, the medium variation is set equal to the standard deviation (SD) value from the literature of 4.3 min [29]. Low, high, and very high variations equal 0.5, 1.5, and 2 times the SD value, respectively. Thus, we can systematically analyze the impacts of varying consultation times on patient waiting times alone and in combination with varying inter-arrival times of patients.

#### 2.3.1. Scenario 1—Fixed Patient Arrivals

Patients’ inter-arrival times are fixed to a constant value in order to separate the effects of variation in consultation times and patient inter-arrival times. It is assumed that patients arrive for their appointments on time every 7.6 min. The first patient arrives and is in the consultation immediately. The consultation takes 7.6 min with variations of 2.15, 4.3, 6.45 or 8.6 min (standard deviation (SD)). Independently of the ongoing consultation’s duration, the second patient arrives after 7.6 min and needs to wait in case a random value corresponding to the SD is added. After the first patient has left, the second patient directly enters the consultation, which also takes 7.6 min plus or minus one of the random values with the defined SD. The third patient arrives 7.6 min after the second patient’s arrival time and needs to wait if any additional variation occurred before. This process is repeated for all following patients.

#### 2.3.2. Scenario 2—Random Patient Arrivals

Both patient arrivals and physician consultation times are varied in order to show the combined effects. This corresponds to a situation with only walk-ins. Patients still arrive every 7.6 min on average, but may also arrive later or earlier. This scenario follows the same process as that of scenario 1 and includes a variation in patient arrival times such that patients arrive every 7.6 min with variations (standard deviation (SD)).

### 2.4. Detailed Half-Day Analyses Based on Scenario 2

To provide further insights, especially on the maximum waiting times, we aimed to analyze the effect of a considerable individual delay in consultation times on the patients’ waiting times. Such extensive consultation times can occur due to unforeseen urgent patient demands due to medical emergencies [30]. To illustrate how a significant delay in patients’ consultations affects the individual patients’ waiting times if there is no reaction, we consider a busy 5 h morning session. We follow the assumptions of scenario 2 with walk-ins and stochastic consultation times. A total of 49 patients arrive during the 5 h session, and the physician consultation time is assumed to be 7.6 min on average, with a standard deviation of 4.3 min.

To analyze the total savings in patients’ waiting times, we exemplify two possible consultation strategies for reacting to unforeseen long consultation times with subsequently long waiting times. Strategy 1 shortens the consultations of all subsequent patients to a fixed 6 min slot. This may be possible if, for example, buffers are implemented in the appointment schedule or the physician can focus on the central patient concerns while rescheduling an appointment on a different day for additional but not urgent demands. Strategy 2 illustrates considerably shorter consultation times for the following five patients to compensate for the delay and resumes the stochastic 7.6 min consultation time afterward.

## 3. Results

In the following, we present the simulation results to quantify the effects of varying physician consultation and patient arrival times on patient waiting times. Based on the results of both scenarios, ‘take-home messages’ are outlined.

### 3.1. Scenario 1: Fixed Patient Arrivals

In the first scenario, patients’ inter-arrival times are fixed to a constant value to separate the effects of variations in consultation times and patient inter-arrival times. It is assumed that all patients with an appointment follow the schedule. As described, the patient inter-arrival times were set to 7.6 min, which is equal to the average physician consultation time in Germany. Thus, it is assumed that patients arrive at their appointments on time every 7.6 min.

Since the average physician’s consultation time equals the patient’s inter-arrival times, no waiting times occur when there are no variations in consultation times. As detailed in Figure 2, greater variations in consultation time, as indicated by the standard deviations in parentheses, lead to longer waiting times for individual patients’ average (blue diamond) and maximum waiting times, although the mean consultation duration remains unchanged at 7.6 min. Further, while average waiting times moderately increase for greater variations in treatment times, the waiting times of individual patients can dramatically increase—up to more than 200 min in this case.

**Take-Home Message 1.** Less variation in consultation times during a day, e.g., by categorizing appointment types and lengths, reduces waiting times. For example, patients with known higher needs can be booked in two (or more) standard slots rather than one slot.

### 3.2. Scenario 2: Random Patients’ Arrival

In the second scenario, both patient arrival and physician consultation times are varied to show the combined effects. This corresponds to a situation with only walk-ins. Patients now arrive, on average, every 7.6 min, but may also arrive later or earlier, and the same assumptions as those detailed above are applied. Figure 3 shows the combined effect of walk-ins and variations in consultation times on the average patient waiting times compared to the average waiting time in scenario 1. If patients’ arrivals are not scheduled, an average waiting time of about 20 min per patient results, even when there is no variation in consultation times. Higher standard deviations in consultation times lead to even longer average waiting times.

**Take-Home Message 2.** Influencing patient arrival behavior, e.g., by scheduling appointments in combination with the respective information of patients, reduces patients’ waiting times.

### 3.3. Detailed Half-Day Consideration: Effects of Consultation Time Strategies on Patients’ Waiting Times

Following the assumptions of scenario 2, the half-day consideration examines 5 h of a working day with stochastic (random) patient arrival and stochastic consultation times. For illustrative purposes, we selected a scenario where a major delay occurred. On this specific day, the effects of two intervention strategies are demonstrated. Such delays may also occur in practice when emergencies have to be interposed. It is straightforward that the earlier these extreme exceedances occur, the greater the impact on the waiting times for the entire day will be if no adjustments are made. Thus, it is essential to compensate them with either constant or reduced consultation times (or both) for as many patients as possible. Figure 4a,b illustrate these options.

#### 3.3.1. Baseline Case

In the baseline case, consultation durations remain unchanged, while a delay occurs for patients 14 and 15. The gray–blue line in Figure 4a indicates an increasing waiting time for patients throughout the subsequent morning. Especially after patient 21, waiting times of more than 60 min occur. In total, this represents the baseline case with a total physician consultation time of about 380 min and a total waiting time of the patients of about 2790 min.

#### 3.3.2. Strategy 1: Fixed, Slightly Reduced Consultation Times without Any Variance

In reaction to the delay in our example, the consultation times are reduced from an average of 7.6 min to a constant 6 min. Hence, any variance in consultation is eliminated, and the physician’s total consultation time is reduced by around 40 min. The dark blue line represents this extreme strategy in Figure 4a. As a result, some patients are treated longer and others are treated for a considerably shorter time. The variations in waiting times result from the variations in patients’ inter-arrival times. The reduced waiting times per patient are shown in Figure 4b. In total, 834 min of patient waiting times can be saved compared to the baseline case by following this intervention strategy. 

#### 3.3.3. Strategy 2: Considerably Shorter Consultation Times for a Few Subsequent Patients

An alternative strategy is to considerably shorten the consultation time for the five subsequent patients to compensate for the unforeseen long consultation time of an additional 15 min. In this example, the physician reduces the following five consultation times by 3 min each. This reduces the total consultation times for these patients by 15 min. All other consultation times remain the same. Thus, the delay does not influence the subsequent patient appointments. Thereby, the physician reduces the patients’ waiting times by 405 min in total (Strategy 2) compared to the baseline case, with no change in consultation durations. The bars in Figure 4b show the total waiting time savings for individual patients compared to the baseline case if the physician follows Strategy 1 or Strategy 2. Due to the cumulative effect of waiting time savings, the total waiting time of patients can be markedly reduced by a fast reaction of the physician. Strategy 1 results in higher waiting time savings for each individual patient compared to Strategy 2 and, thus, results in higher total savings. However, the efficiency of Strategy 2 can be considered higher compared to Strategy 1 in terms of the necessary reduction in consultation times. We calculated the average waiting time savings per minute of reduced consultation time to determine the efficiency. For this purpose, the ratio of total waiting time savings divided by the reduction of total consultation times is determined. This ratio equals 21 for Strategy 1 and 27 for Strategy 2. In other words, Strategy 1 and Strategy 2 result in a reduction in average waiting times of 21 and 27 min, respectively, for each minute of reduced consultation time.

**Take-Home Message 3.** The sooner a delay in consultation time is detected and the sooner the physician is able to intervene with shorter and less variable consultation times for as many patients as possible, the greater the potential reduction of waiting times for all subsequent patients will be. In practice, this can be achieved, e.g., by focusing on the most clinically important patient need(s) with rescheduling of the patients for a different day to address other needs.

## 4. Discussion

Drawing on established operations research principles and methods, our simulation analyses aim at increasing physicians’ and other practice personnel´s awareness of the key factors that drive patients’ waiting times, namely, patient inflow and consultation length. Although most assumptions are based on real-life data, the results are not ready-to-use-instructions, but they exemplify important principles that allow for the development of long- and short-term implementation strategies for individual practice settings.

In the framework of intraday scheduling, we focused on the variability of direct waiting times with three take-home messages for general practices. The first message (fewer variations in consultation times lead to shorter waiting times) emphasizes that measuring, categorizing, and determining consultation times is needed to derive practice-specific standard (fixed) appointment lengths as the basis for improved operational workflows, e.g., consecutively scheduling predictable appointments as check-ups or follow-ups. For implementation, the variety of appointment types has to be reduced, consultation types have to be categorized, contingency plans for unpredicted events have to be developed, and scarce resources have to be increased [30]. However, single patients’ needs are hardly predictable when scheduling, since a variety of patients and health issues is a characteristic of family medicine [31]. It is a known challenge that patients often do not indicate all consultation reasons to the receptionist. This also implies that patients with known higher needs may require a double or triple slot of the practice’s standard consultation time, which can be considered when scheduling. From our own experience and discussions with general practitioners (GPs) and practice assistants, this approach is realistic because the personnel typically knows which patients require more physician time.

The second take-home message (appointment scheduling reduces patients’ waiting times) points at the need for scheduling systems rather than walk-ins, although some patient populations, e.g., younger patients [32] and those with difficulties in contacting practices in advance, prefer same-day appointments. In the COVID-19 pandemic, the issue of appointment scheduling has become essential to prevent disease transmission within practices, e.g., with separate office hours for suspected or diagnosed COVID-19 cases [33]. The so-called “advanced access scheduling”, which is one of the most investigated models, consists of a combination: It keeps a defined number of same-day appointment slots unscheduled to preserve them for acute care and unforeseen events [34], while foreseeable patients, e.g., for chronic care, are still prescheduled [35]. A recent literature review identified the five pillars for the advanced access model. One was described as the processes of appointment booking and scheduling [36]. Reducing waiting times and no-show rates can increase the patient volume and the productivity of the provider [37]. However, it does not provide a benefit if the patient demand for appointments is constantly higher than the physicians’ capacity [38]. This advanced scheduling model is quite similar to the carve-out model, which reserves fixed time slots for physicians to take care of urgent demands [30]. These fixed time slots are unoccupied and act as a buffer to offset urgent demands and/or delays. According to the so-called stochastic carve-out model, a few open slots at the beginning of the session and during the day, e.g., every hour, can reduce waiting times [38]. The ORCA (Optimal Reservation of Capacity for Appointments) optimization model provides strategies for inter-day appointment scheduling, i.e., booking of foreseeable appointments on days that typically have less demand. It aims at determining physicians’ optimal appointment capacities to reduce waiting times while balancing the providers´ workload over the week and preventing overtime [20]. Important research has pointed at the relationship between physician idle time and patients´ waiting time; to minimize waiting times, the physician idle time needs to increase [16]. For implementation in real life, this can be done by adding some idle time to an average appointment length, which—if not needed to reduce waiting times—can be used by physicians for other tasks, e.g., office work.

The third take-home message (the need for early interventions to counteract consultation delays) shows the effects of strategies for acute interventions. Patients with predictable requests, e.g., check-up or follow-up consultations, can be postponed, and tasks that can be delegated to non-physician personnel can save time during the actual consultation [34]. Various delegation models, e.g., for wound care, hypertension management, and other health topics, are evaluated or even routinely implemented in various countries [39,40] to increase physicians’ capacities. In addition, alternative care strategies in primary care settings were applied in different studies:

A study among pediatric patients showed that medical assistant service applications on smartphones improved the follow-up attendance [41]. Online pre-registration was shown to reduce waiting times in walk-in patients in an outpatient medical department [42]. In addition, shared medical appointments and group office visit approaches are innovative approaches for increasing the capacity of general practices and reducing waiting times [43,44].

Furthermore, phone, email, and video consultations are effective ways to reduce waiting times [34]. Their use increased during the COVID-19 pandemic [45,46], and they found a broad acceptance among German medical professionals [47]. They were especially helpful for preventing the transmission of COVID-19 while keeping up high medical care [47].

Simulation approaches allow for investigations of alternative scenarios without cost-intensively modifying real systems [23]. To implement insights into general practices, a number of steps are needed, which start from an understanding of the real-life setting and the communication processes and decisions of physicians and their teams. Furthermore, the implementation of a scheduling system requires ongoing communication with patients, e.g., to reduce walk-ins and achieve the punctuality needed to successfully realize a selected approach.

## Strengths and Limitations

The simulation model, with its focus on two variables—patient arrival and physician capacity—reduces the complexity of scheduling in real life to better illustrate key parameters, but this implies a simplification. The average consultation time of 7.6 min [29] does not cover any additional tasks of the physician, such as documentation and other patient-related inquiries. However, a comparison of publicly available data on waiting times [8] with the results of the simulations showed a plausible agreement. Comparable simulation studies can be performed to investigate a deviating patient volume with adjusted treatment durations, e.g., during flu seasons. The model used does not reflect other scheduling options, such as buffer slots and inter-day appointments, varying patient flows depending on the weekday [20], no-shows and walk-ins, or differing practice structures, e.g., group practices. We did not consider alternative strategies, e.g., delegation models or smartphone applications for appointment scheduling. In addition, some of the strategies evaluated may not necessarily be feasible for each practice setting.

## 5. Conclusions

This paper illustrates the large effects of small changes on waiting times and aims to increase the awareness of this topic based on typical scenarios. The results are not generally applicable solutions, but illustrate potential applications of these principles to practices’ workflows. Prospectively, extending the simulation model beyond the two key variables—consultation times and patient inter-arrival times—will lead to more complex but also more accurate representations of real-world practices. Given the physician shortages in many countries, there is a need to intensify empirical research on workflows in primary care and on how to implement findings from theoretical frameworks to better support general practitioners and their teams as the largest ambulatory health care profession.

## Figures and Tables

**Figure 1 ijerph-20-01767-f001:**
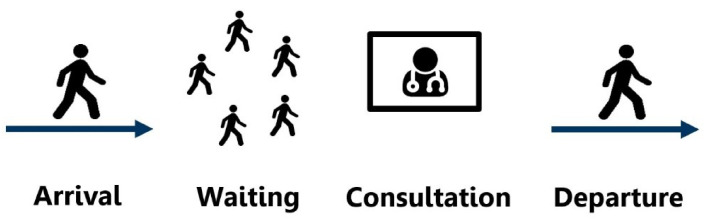
A queuing system in a general practice has three key components: arrival process, services (physician consultation and idle time), and queuing discipline (treatment order, e.g., as scheduled with/without emergency patients).

**Figure 2 ijerph-20-01767-f002:**
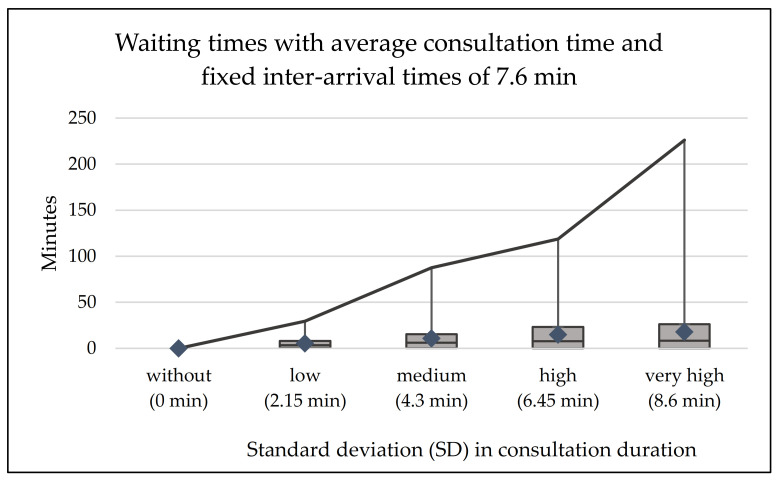
Average, 25%, 75%, and maximum patient waiting times for various standard deviations (in parentheses) in consultation times if only patients with appointments arrive on time every 7.6 min.

**Figure 3 ijerph-20-01767-f003:**
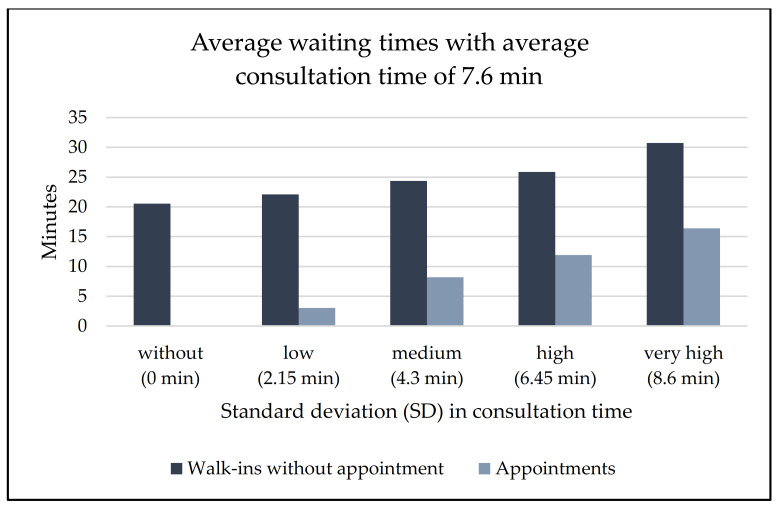
Comparison of patient waiting times in scenario 1 (‘fixed appointments’) and scenario 2 (‘walk-ins’) for different standard deviations in consultation time.

**Figure 4 ijerph-20-01767-f004:**
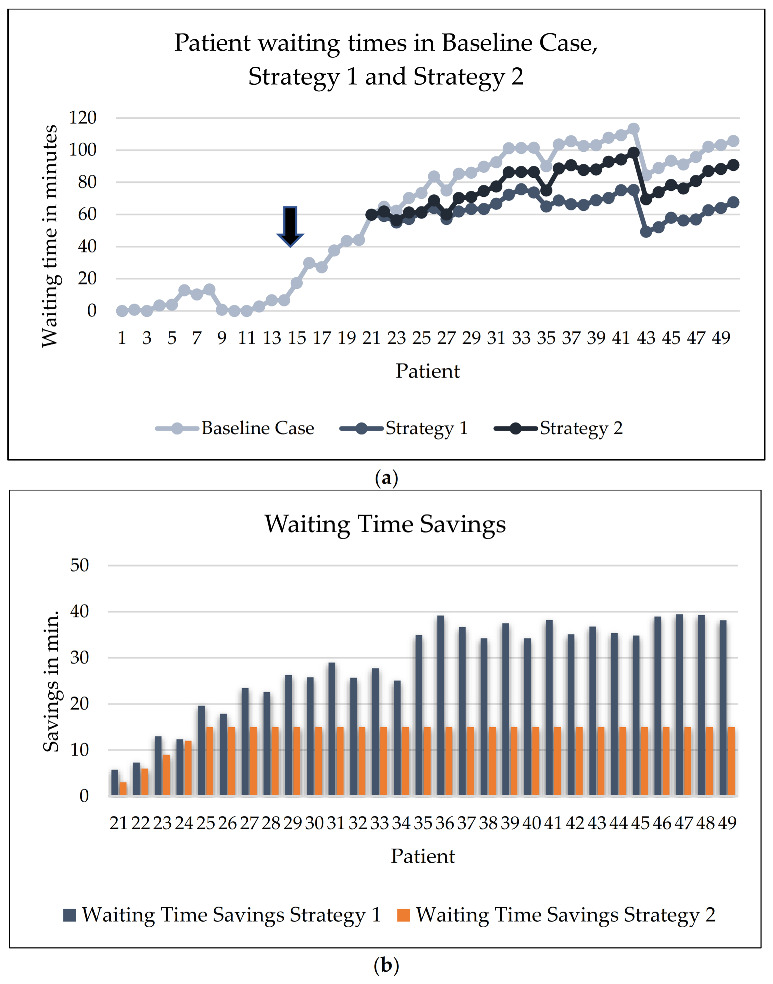
(**a**) Scenario 2 with an unforeseen long consultation time of an additional 15 min for patients 14 and 15 (see arrow). The light blue line shows the increase in waiting time if the treatment duration remains unchanged (Baseline). The mid-blue line shows the reduction in waiting times for Strategy 1 (reduction in consultation time to constantly 6 min), and the dark blue line shows the reduction in waiting times for Strategy 2 (reduction in consultation time by 3 min for the following five patients). (**b**) Waiting time savings for all subsequent patients when following Strategy 1 or Strategy 2 compared to the baseline case.

## Data Availability

The datasets used and analyzed are available from Prof. Werners, Institute of Management, Operations Research, Ruhr University Bochum by request.

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
