# Peer review of "Small Changes in Patient Arrival and Consultation Times Have Large Effects on Patients’ Waiting Times: Simulation Analyses for Primary Care"

_ijerph, 2023, doi:10.3390/ijerph20031767_

Round 1

Reviewer 1 Report

This is a novel study that applies operations research approaches to the case of wait times at a physician practice. Overall, I think that the topic is of significant importance, the methods are solid, and the manuscript is well-written. I have a few comments:

1.      Page 4 Line 158: in this half-day analysis, the authors simulate a 5-hour morning session, but why does total consultation time in the results section on Page 6 consistently exceed 300 minutes? Also, I think it would make more sense to simulate for the entire day, because the authors assume that patients that arrive on a day must be seen on the same day. Any events in the morning will continue to affect wait times in the afternoon.

2.      Figure 2: In addition to the mean and max, it would be helpful to plot or show 25th percentile, median, and 75th percentile to illustrate the distribution of wait times.

3.      Unlike Figures 2 and 3, for the half-day analysis, there’s only one line for each scenario (Figure 4a). What standard deviation in consultation time did the authors pick?

4.      Regarding Strategy 2 (Section 3.2.3), why didn’t the authors also include a line for it in Figure 4a? Also, why does Figure 4b start from Patient 21. Patients 16-20 should also see wait times.

5.      Page 7 Line 261: the authors state that “Strategy 2 resulted in a higher waiting time saving earlier than Strategy 1”. However, Figure 4b seems to show that wait time savings from Strategy 1 are always greater than from Strategy 2.

6.      This is a very interesting topic, and as the authors noted, they have simplified the real-world situations in this paper. It would be great for them to extend their simulation to account for more complex scenarios. For example, we know that physicians see more patients during the flu season. Suppose a physician sticks with Strategy 2 to handle delays, how do wait time estimates change in the flu season and other times of a year?

Author Response

Reviewer 1:

This is a novel study that applies operations research approaches to the case of wait times at a physician practice. Overall, I think that the topic is of significant importance, the methods are solid, and the manuscript is well-written. I have a few comments:

  1. Page 4 Line 158: in this half-day analysis, the authors simulate a 5-hour morning session, but why does total consultation time in the results section on Page 6 consistently exceed 300 minutes? Also, I think it would make more sense to simulate for the entire day, because the authors assume that patients that arrive on a day must be seen on the same day. Any events in the morning will continue to affect wait times in the afternoon.

Authors’ response: Thank you very much for the valuable comment. The consultation time exceeds 300 minutes due to the variations in patient arrival and physician consultation times. Additionally, we simulated each day with a two-hour break during lunchtime. Therefore, all afternoon sessions start with no patients in the waiting queue. We clarified this in the manuscript.

Revised text (lines 116-119): On each day, patients are treated for eight hours, separated into a five-hour morning and three-hour afternoon session, interrupted by a two-hour break. When consultation times exceed the five-hour morning session, the physician sees patients during the break and works overtime.

  1. Figure 2: In addition to the mean and max, it would be helpful to plot or show 25th percentile, median, and 75th percentile to illustrate the distribution of wait times.

Authors’ response: Thank you for the valuable remark. We have added the 25th percentile, the median and the 75th percentile to Figure 2. For reasons of conciseness, we have changed the presentation of the figure accordingly.

Revised text (line 191): See Figure 2

  1. Unlike Figures 2 and 3, for the half-day analysis, there’s only one line for each scenario (Figure 4a). What standard deviation in consultation time did the authors pick?

Authors’ response: Thank you for the comment. We chose a standard deviation of 4.3 minutes that is found in the literature for physician consultation times. We revised the text accordingly.

Revised text (lines 167-169): 49 patients arrive during the 5-hour-session, and the physician consultation time is assumed to be 7.6 minutes on average with a standard deviation of 4.3 minutes.

  1. Regarding Strategy 2 (Section 3.2.3), why didn’t the authors also include a line for it in Figure 4a? Also, why does Figure 4b start from Patient 21. Patients 16-20 should also see wait times.

Authors’ response: Thank you very much for the valuable remark. We added a line for the individual patient waiting times for Strategy 2 in Figure 4a. Figure 4b now starts from patient 22 since it shows the waiting time savings for each patient which does not mean that earlier patients do not have waiting times. Thus, it compares the waiting times from the baseline with both Strategies, respectively. Since the physician in this example reacts with a strategy at patient 21 there are no waiting time savings for patients 1-21 compared to the baseline.

Revised text (line 258): See Figure 4a

  1. Page 7 Line 261: the authors state that “Strategy 2 resulted in a higher waiting time saving earlier than Strategy 1”. However, Figure 4b seems to show that wait time savings from Strategy 1 are always greater than from Strategy 2.

Authors’ response: Thank you very much for the important remark. We agree that this sentence does not fit the current numerical example. We corrected the statement accordingly.

Revised text (lines 283-286): Strategy 1 results in higher waiting time savings for each individual patient compared to Strategy 2 and thus results in higher total savings. However, the efficiency of Strategy 2 can be considered higher compared to Strategy 1 in terms of necessary consultation times reduction.

  1. This is a very interesting topic, and as the authors noted, they have simplified the real-world situations in this paper. It would be great for them to extend their simulation to account for more complex scenarios. For example, we know that physicians see more patients during the flu season. Suppose a physician sticks with Strategy 2 to handle delays, how do wait time estimates change in the flu season and other times of a year?

Authors’ response: Thank you very much for the comment. We agree that more complex situations occur. However, a more complex model formulation is required that goes beyond the scope of this paper. This paper aims to raise awareness for strategies to improve workflows in general practices. Therefore, we focus on simple quantitative examples that highlight the importance of constantly reviewing the given structures in general practices. Based on this, more detailed research can be conducted to determine the best opportunities to improve individual general practices' workflows.

Revised text (lines 384-386): Comparable simulation studies can be performed to investigate a deviating patient volume with adjusted treatment durations, e.g., during flu seasons.

Reviewer 2 Report

I would like to thank the editors of this journal for the opportunity to give my point of view on this interesting research project. On the other hand, I would like to congratulate the authors of this research, I believe that it is very necessary to investigate the processes of patient care and how to intervene in the administrative methodology of the centre to reduce the waiting time of the patient and make their experience more pleasant, as well as to better manage the time of doctors to reduce stress and increase their effectiveness.

I believe that the study is well planned and the methodology is appropriate, but I would like to make some comments that I think could be considered by the authors in order to improve some aspects of the study.

I think it would be appropriate to include in the introduction and discussion the possibility offered by smartphone applications to manage appointments with patients, could this influence your study? Considering that access to these apps may be limited, would an app that allows the patient to see in real time the attention of the other patients on their list and thus be able to manage their stay in the waiting room be useful? Here are some reference studies.

https://doi.org/10.1097/jcn.0b013e3181b8e82e
https://pubmed.ncbi.nlm.nih.gov/27442205/
https://doi.org/10.1007/s00417-018-4080-z

In my opinion, another concept that could be improved is the assessment of the results within the results section. This is an aspect of form and I believe that it does not affect the reading and correct evaluation of the paper, but normally, the evaluation of the results is made in the discussion and not in the results section itself.

I believe that the possibility of patient time management with a smartphone application should be included in the limitations of the study, or at least contemplated as a future study.

Once again, I congratulate the researchers for the effort and dedication they have put into this great study and I thank the journal editors for the opportunity to have contributed my opinion on the research.

Author Response

Reviewer 2:

I would like to thank the editors of this journal for the opportunity to give my point of view on this interesting research project. On the other hand, I would like to congratulate the authors of this research, I believe that it is very necessary to investigate the processes of patient care and how to intervene in the administrative methodology of the centre to reduce the waiting time of the patient and make their experience more pleasant, as well as to better manage the time of doctors to reduce stress and increase their effectiveness. I believe that the study is well planned and the methodology is appropriate, but I would like to make some comments that I think could be considered by the authors in order to improve some aspects of the study.

  1. I think it would be appropriate to include in the introduction and discussion the possibility offered by smartphone applications to manage appointments with patients, could this influence your study? Considering that access to these apps may be limited, would an app that allows the patient to see in real time the attention of the other patients on their list and thus be able to manage their stay in the waiting room be useful? Here are some reference studies.

https://doi.org/10.1097/jcn.0b013e3181b8e82e
https://pubmed.ncbi.nlm.nih.gov/27442205/
https://doi.org/10.1007/s00417-018-4080-z

Authors’ response: Thank you very much for your helpful suggestions.

Revised text in discussion (lines 358-365): In addition, alternative care strategies in primary care settings were applied in different studies:

A study among pediatric patients showed that medical assistant service ap-plications on smartphones improved the follow-up attendance [44]. Online pre-registration was shown to reduce waiting times in walk-in patients in an outpatient medical department [47]. In addition, shared medical appointments and group office visit approaches are innovative approaches to increase the capacity of general practices and reduce waiting times [45, 46].

  1. In my opinion, another concept that could be improved is the assessment of the results within the results section. This is an aspect of form and I believe that it does not affect the reading and correct evaluation of the paper, but normally, the evaluation of the results is made in the discussion and not in the results section itself.

Authors’ response: Thank you very much for your comment. We chose this format intentionally to provide practical implications directly when presenting the theoretical results.

  1. I believe that the possibility of patient time management with a smartphone application should be included in the limitations of the study, or at least contemplated as a future study.

Authors’ response: Thank you for this valuable comment. We included this information in the discussion and limitation sections.

Revised text in Strengths and Limitations (lines 389-390): We did not consider alternative strategies, e.g. delegation models or smartphone applications for appointment scheduling.

Once again, I congratulate the researchers for the effort and dedication they have put into this great study and I thank the journal editors for the opportunity to have contributed my opinion on the research.

Reviewer 3 Report

The authors present a simulation analyzes for primary care with the aim of providing useful strategies to decrease direct waiting time, which represent a patient's time in the practice waiting for the consultation, and which is associated with less patient satisfaction and medical compliance as well as increased no-show rates. This issue is of great importance in primary care, where the average consultation time per patient is difficult to predict, given the wide range of problems for which a general practitioner is seen.

I have carefully reviewed the manuscript and I think that the work is of interest to IJERPH readers, and meets the quality conditions necessary for its publication, although they might appreciate the following comments.

This work is aimed at primary care professionals and as is well known, and the authors point out, high direct waiting times are associated with less patient satisfaction as well as increased no-show rates. In the first paragraph of the Discussion they indicate "our simulation analyzes aim at increasing physicians' and other practice personnel's awareness for the key factors that drive patients' waiting times, namely patient inflow and consultation length". These key factors are well known by general practitioners, and we also know that frequently the single patient's needs are hardly predictable when scheduling, except for administrative issues, collection of laboratory tests, renewal of prescriptions or review consultations. But on many occasions, the patient can request a consultation without indicating the real reason to the receptionist, or there may be an important reason (a psychosocial problem, for example) that must be addressed without delay, even if this means causing a significant delay in the entry of the following patients.

To reduce the negative effects of direct wait times, at least in patient satisfaction, a simple strategy is to place a sign in the waiting room indicating that the appointment time is approximate. On the door of my practice there is one that indicates that -"The time of the appointments is approximate. Each patient needs a consultation time that is difficult to predict”.

I think that the authors could make a more explicit comment on this peculiarity of family medicine in Discussion.

Author Response

Reviewer 3:

The authors present a simulation analyzes for primary care with the aim of providing useful strategies to decrease direct waiting time, which represent a patient's time in the practice waiting for the consultation, and which is associated with less patient satisfaction and medical compliance as well as increased no-show rates. This issue is of great importance in primary care, where the average consultation time per patient is difficult to predict, given the wide range of problems for which a general practitioner is seen. I have carefully reviewed the manuscript and I think that the work is of interest to IJERPH readers, and meets the quality conditions necessary for its publication, although they might appreciate the following comments.

  1. This work is aimed at primary care professionals and as is well known, and the authors point out, high direct waiting times are associated with less patient satisfaction as well as increased no-show rates. In the first paragraph of the Discussion they indicate "our simulation analyzes aim at increasing physicians' and other practice personnel's awareness for the key factors that drive patients' waiting times, namely patient inflow and consultation length". These key factors are well known by general practitioners, and we also know that frequently the single patient's needs are hardly predictable when scheduling, except for administrative issues, collection of laboratory tests, renewal of prescriptions or review consultations. But on many occasions, the patient can request a consultation without indicating the real reason to the receptionist, or there may be an important reason (a psychosocial problem, for example) that must be addressed without delay, even if this means causing a significant delay in the entry of the following patients.

Authors’ response: Thank you very much outlining the differences in patients’ needs. We adapted the text accordingly.

Revised text in discussion (lines 315-318): However, single patient’s needs are hardly predictable when scheduling since a variety of patients and health issues are a characteristic of family medicine [43]. It is a known challenge, that patients often do not indicate all consultation reasons to the receptionist.

  1. To reduce the negative effects of direct wait times, at least in patient satisfaction, a simple strategy is to place a sign in the waiting room indicating that the appointment time is approximate. On the door of my practice there is one that indicates that -"The time of the appointments is approximate. Each patient needs a consultation time that is difficult to predict”.

Authors’ response: Thank you very much for sharing your approach with us.

  1. I think that the authors could make a more explicit comment on this peculiarity of family medicine in Discussion.

Authors’ response: Thank you very much for this recommendation. We pointed it out more clearly.

Revised text in discussion (lines 315-318): However, single patient’s needs are hardly predictable when scheduling since a variety of patients and health issues are a characteristic of family medicine [43]. It is a known challenge, that patients often do not indicate all consultation reasons to the receptionist.

Round 2

Reviewer 1 Report

The authors have done a great job addressing my comments, and the manuscript has improved significantly.